# Full-Waveform LiDAR Point Clouds Classification Based on Wavelet Support Vector Machine and Ensemble Learning

**DOI:** 10.3390/s19143191

**Published:** 2019-07-19

**Authors:** Xudong Lai, Yifei Yuan, Yongxu Li, Mingwei Wang

**Affiliations:** 1School of Remote Sensing and Information Engineering, Wuhan University, Wuhan 430079, China; 2Key Laboratory for National Geographic Census and Monitoring, National Administration of Surveying, Mapping and Geoinformation, Wuhan 430079, China; 3Beijing Key Laboratory of Urban Spatial Information Engineering, Beijing 100038, China; 4Institute of Geological Survey, China University of Geosciences, Wuhan 430074, China

**Keywords:** full-waveform LiDAR, point cloud classification, support vector machine, wavelet kernel function, ensemble learning

## Abstract

Light Detection and Ranging (LiDAR) produces 3D point clouds that describe ground objects, and has been used to make object interpretation in many cases. However, traditional LiDAR only records discrete echo signals and provides limited feature parameters of point clouds, while full-waveform LiDAR (FWL) records the backscattered echo in the form of a waveform, which provides more echo information. With the development of machine learning, support vector machine (SVM) is one of the commonly used classifiers to deal with high dimensional data via small amount of samples. Ensemble learning, which combines a set of base classifiers to determine the output result, is presented and SVM ensemble is used to improve the discrimination ability, owing to small differences in features between different types of data. In addition, previous kernel functions of SVM usually cause under-fitting or over-fitting that decreases the generalization performance. Hence, a series of kernel functions based on wavelet analysis are used to construct different wavelet SVMs (WSVMs) that improve the heterogeneity of ensemble system. Meanwhile, the parameters of SVM have a significant influence on the classification result. Therefore, in this paper, FWL point clouds are classified by WSVM ensemble and particle swarm optimization is used to find the optimal parameters of WSVM. Experimental results illustrate that the proposed method is robust and effective, and it is applicable to some practical work.

## 1. Introduction

The remote sensing technique is based on the sensor recorded energy that is reflected or emitted from the Earth’s surface [1]. Therefore, remote sensing can acquire information about ground objects without physical contact and can be divided into two categories: passive and active. Passive remote sensing data mostly exist in the form of spectral images, which makes it difficult for them to describe 3D space features and they usually require geometric correction.

Light Detection and Ranging (LiDAR) is a type of active remote sensing technique. Different from the 2D information displayed by spectral images, the advantage of LiDAR is to directly generate point clouds with large area coverage and high-precision coordinates [2]. In most cases, LiDAR point clouds are used to produce elevation data, such as the Digital Elevation Model (DEM) and Digital Surface Model (DSM), which is typically employed as ancillary information to assist passive remote sensed data in classification [3,4,5,6]. With the hardware equipment of LiDAR, more and more features are extracted for point cloud classification. According to the necessity of training samples, classification methods can be summarized as two types: unsupervised and supervised. Unsupervised methods are based on a suitable definition of similarity between data without any prior knowledge, and have been employed in the field of LiDAR point cloud classification [7], including K-means, iterative self-organizing data analysis (ISODATA), fuzzy c-means (FCM) [8,9,10], etc. However, the classification results may not correspond to the classes of ground objects and the classification accuracy of these methods has difficulty meeting practical requirements in some situations. Supervised methods, such as decision tree, random forest (RF), artificial neural network (ANN), and k-nearest neighbors (KNN) algorithm, have also been applied in the field of LiDAR point cloud classification [11,12,13,14]. They define class labels with prior knowledge deduced from training samples and the classification accuracy is improved by repeatedly modifying the training samples. However, these methods may have difficulty solving the classification problems with a small number of samples and are sensitive to feature correlation.

Different from traditional LiDAR, full-waveform LiDAR (FWL) is a new type, which records the backscattered echo in very small intervals and obtains a continuous echo waveform [15]. It is able to obtain more footprints, echoes and the actual situation of data collection. By decomposing full-waveform data, point clouds are calculated and waveform features such as the amplitude, width, peak location and intervals of peak location are extracted. The classification of LiDAR point clouds determines their categories and the commonly used features are intensity, geometry, texture and elevation. For FWL, except for the conventional features above, waveform features can be used in the classification process of point clouds, and the methods mentioned above are also applicable for FWL point cloud classification [16,17,18]. Moreover, the affiliation of waveform features in convention is helpful to improve the classification accuracy [17,19].

As a supervised method in machine learning, support vector machine (SVM) shows potential for effective and efficient classification of different types of data, and has a strong ability to solve a series of problems [20,21,22]. Because of its optimal hyper-plane in feature space based on structural risk minimization theory, SVM has become a widely used classification tool in need of a small amount of training data and fewer computational efforts [23,24,25], and it has been utilized in the field of FWL point cloud classification [26,27]. In SVM, the kernel function is an important factor affecting the classification results, while the commonly used liner, polynomial and sigmoid kernel functions may easily cause over-fitting or under-fitting in the lack of localization ability. Wavelet analysis is a powerful estimation technique for the time–frequency analysis of a signal [28], representing a signal by different resolutions. Therefore, wavelet functions, which have a good localization property, are used to build kernel functions employed in the construction of wavelet SVM (WSVM). In addition, the penalty factor and kernel function parameter play an important role in the classification result via SVM. To obtain better classification accuracy, parameter optimization is necessary, whose essence is a combinatorial optimization problem, and swarm intelligence algorithm can be used to solve this kind of problem because of its ability to obtain a satisfactory solution.

Moreover, under some circumstances, a single classifier is difficult to solve the case when the differences between feature values are very small and it may not label data correctly. In this case, ensemble learning is a promising research direction, and it is composed of a set of individual component classifiers (base classifiers) whose predictions are combined to determine the final results, improving the overall classification accuracy [29,30]. When SVMs are used as base classifiers, the ensemble system is called the SVM ensemble, which has been applied in many fields, such as hyperspectral data classification, dynamic financial distress prediction, credit card evaluation [31,32,33], etc. An ensemble learning system with different base classifiers highlights the heterogeneity, but ignores the homogeneity, while a system with the same base classifiers only focuses on the homogeneity and neglects the heterogeneity. In such a system based on WSVM, the base classifiers are composed of SVMs using wavelet kernel functions, which are in different forms but all considered on the basis of wavelet analysis and theoretically related. Therefore, the classification process for FWL point clouds are completed by WSVM ensemble in this paper, employing five variable wavelet kernel functions, and each classifier constructed by them, respectively, is used twice. At the same time, particle swarm optimization (PSO), which is a swarm intelligence algorithm with a good performance and stable convergence to the optimal solution, is utilized to optimize the SVM parameters.

## 2. Feature Extraction for FWL

According to the type of device, LiDAR point cloud classification can be conducted on the basis of many features, such as intensity, geometric, textural, and multispectral features. Compared with traditional LiDAR, FWL is able to extract more points, showing more details on structure. Therefore, geometric features are adopted to show the positional relationship between point clouds. However, it reflects the characteristics of discrete point clouds and neglects the interaction of the entire laser pulse with the ground objects. Hence, waveform features can be seen as a powerful complement to the classification process. For FWL, waveform decomposition of original data, whose essence is to fit the waveform with a function, is a useful technique to extract data information and should be performed before the process of classification. In this paper, before waveform decomposition, filtration of noise in the raw data [34] is conducted to reduce the noise interference for classification results, and then a set of Gaussian functions [35] is employed to fit the waveform. The relevant parameters of the function thus obtained are called waveform features of point clouds, reflecting the position and backscattering property of the targets during laser transmission.

Geometric features chosen for classification include the elevation (*h*), elevation standard deviation ($\sigma_{h}$), volume density (ρ), curve (*c*), vertical angle ($N_{Z}$), and vertical angle variance ($\sigma_{N_{Z}}$). Before the extraction of geometric features, a restricted 3D neighborhood ω is defined. A covariance matrix of 3D coordinates and its eigenvalues $\lambda_{1}$, $\lambda_{2}$, and $\lambda_{3}$ ($\lambda_{1}<\lambda_{2}<\lambda_{3}$) are computed on the basis of 3D points in ω. The corresponding eigenvector of $\lambda_{1}$ is the normal vector ($N_{X}$, $N_{Y}$, $N_{Z}$). Geometric features take advantage of the geometric shape intuitively and show geometric relationships between point clouds. However, these features cannot give the radiometric information expressing the reflectance properties of ground objects. While waveform features provide both geometric information and radiometric information, which is used to distinguish different surface materials. Furthermore, compared with the geometric features extracted from a specific neighborhood, the waveform features can reflect the characteristics of the pulse emission direction directly.

Waveform features extracted in this paper are the echo amplitude (*A*), width (σ), peak location (*u*) and intensity (*I*). Amplitude is the peak value of the echo signal, width is the standard deviation of each Gaussian component, peak location is the maximum position of the Gaussian function and intensity is the backscattered energy of a single Gaussian component, which is calculated by integrating each Gaussian component. All used features and their explanations are shown in Table 1. [19].

## 3. Methodology of LiDAR Point Clouds Classification

### 3.1. Construction of WSVM Model

SVM is a machine learning method which classifies data by creating a hyper-plane. Assume a set of training data $X=\{\left(x_{1},y_{1}\right),\left(x_{2},y_{2}\right),\ldots ,\left(x_{n},y_{n}\right)\}$, where $X_{i}\in R^{d}$, and $y_{i}\in \left\{-1,+1\right\}$ is a class label. Then, the training process can be transformed into the optimization of the following expressions [36,37]:(1)$$min\phi \left(\omega ,\epsilon \right)=\frac{1}{2}{\left||\omega |\right|}^{2}+C\sum_{i=1}^{n}\epsilon_{i}$$
(2)$$y_{i}\left(\omega^{T}x_{i}+b\right)\leq 1-\epsilon_{i}$$
where ω is the normal vector of the hyper-plane, and $\epsilon_{i}$ ($\epsilon_{i}\leq 0$) is a slack variable used for measuring classification errors. *C* is the penalty parameter, *b* is the bias or threshold, and ϕ is a function that maps the input data to a higher-dimensional space where data can be linearly separable. To avoid high-dimensional calculations, the mapping step can be skipped by kernel function, which is able to calculate the mapping directly. Additionally, the kernel function is defined as follows:(3)$$K\left(x_{i},x_{j}\right)=\phi {\left(x_{i}\right)}^{T}\cdot \phi \left(x_{j}\right)$$

The series of commonly used kernel functions are global functions rather than local functions, however SVM is an approximation model where a local function basis is better than a global basis [38,39]. The essence of wavelet analysis is to approximate signals using a set of functions generated by dilations and translations of a mother wavelet function [40], which means that a good localization property in the time and frequency domain is presented and the details of a signal are extracted [41]. It also has the advantages of a low redundancy, high stability, and well adaptability to high-dimensional data. The construction process of the wavelet kernel function is as follows [42,43]:

The one-dimensional wavelet function can be described as:(4)$$h\left(x\right)=\prod_{i=1}^{d}h\left(x_{i}\right)$$

Then, the dot-product of the wavelet kernel is
(5)$$K\left(x,x^{\prime }\right)=\prod_{i=1}^{d}h\left(\frac{x_{i}-c_{i}}{a}\right)h\left(\frac{x_{i}^{\prime }-c_{i}^{\prime }}{a}\right)$$
where h(x) is a mother wavelet, $x,x^{\prime }\in R^{d}$, *a* is a dilation factor, and *c* is a translation factor. According to the translation invariant kernel theorem $K\left(x,x^{\prime }\right)=K\left(x-x^{\prime }\right)$, Equation (Equation 5) can be transformed as:(6)$$K\left(x,x^{\prime }\right)=\prod_{i=1}^{d}h\left(\frac{x_{i}-x_{i}^{\prime }}{a}\right)$$

Then, the Gaussian, Shannon, Mexican Hat, Morlet, and Harmonic wavelet functions are used as mother wavelets to construct wavelet kernel functions and the wavelet kernel functions are defined as follows [44,45,46]:

Gaussian:(7)$$K\left(x,x^{\prime }\right)=\prod_{i=1}^{d}exp\left[-\frac{{\left(x_{i}-x_{i}^{\prime }\right)}^{2}}{2\sigma^{2}}\right]$$

Shannon:(8)$$K\left(x,x^{\prime }\right)=\prod_{i=1}^{d}\frac{sin(\frac{\pi }{2}\times \frac{x_{i}-x_{i}^{\prime }}{\sigma })}{\frac{\pi }{2}\times \frac{x_{i}-x_{i}^{\prime }}{\sigma }}cos\left(\frac{3\pi }{2}\times \frac{x_{i}-x_{i}^{\prime }}{\sigma }\right)$$

Mexican Hat:(9)$$K\left(x,x^{\prime }\right)=\prod_{i=1}^{d}[1-\frac{{\left(x_{i}-x_{i}^{\prime }\right)}^{2}}{\sigma^{2}}exp\left[-\frac{{\left(x_{i}-x_{i}^{\prime }\right)}^{2}}{2\sigma^{2}}\right]$$

Morlet:(10)$$K\left(x,x^{\prime }\right)=\prod_{i=1}^{d}cos\left(1.75\times \frac{x_{i}-x_{i}^{\prime }}{\sigma }\right)exp\left[-\frac{{\left(x_{i}-x_{i}^{\prime }\right)}^{2}}{2\sigma^{2}}\right]$$

Harmonic:(11)$$K\left(x,x^{\prime }\right)=\prod_{i=1}^{d}\frac{e^{i4\pi \frac{x_{i}-x_{i}^{\prime }}{\sigma }}-e^{i2\pi \frac{x_{i}-x_{i}^{\prime }}{\sigma }}}{i2\pi \frac{x_{i}-x_{i}^{\prime }}{\sigma }}$$

Wavelet kernel functions are able to approximate arbitrary functions with a good localization property and have the ability of multi-scale analysis [47]. They amplify the difference between the values of the samples’ features and improve the stability of classifier model. Moreover, unlike the commonly used kernel functions that are correlative and redundant, the wavelet kernel functions are orthogonal [48]. These advantages mean that WSVM has a better generalization ability, higher accuracy, and lower computational complexity.

### 3.2. Parameter Optimization

SVM parameters have a great influence on the classification results. SVMs constructed by the above five wavelet kernel functions all have the penalty parameter *C* and kernel function parameter σ. *C* controls the generalization capacity of the classifier, while σ determines the distribution of data after mapping it into a new feature space. PSO is a population-based parallel search algorithm using a group of particles. It has been noticed that members of a group seem to share information among them, a fact that leads to increase the efficiency of the current group. A particle moves toward the optimum according to its present velocity, its previous experience, and the experience of its neighbors. In a n-D search space, the position and velocity of the *i*th particle are represented as vectors $X_{i}=x_{i,1},\ldots ,x_{i,n}$ and $V_{i}=v_{i,1},\ldots ,v_{i,n}$, where each element is coded by real values. Let $Pbest_{i}$ and Gbest be the best position of the *i*th particle and the group’s best position thus far, respectively. The velocity and position of each particle are updated as follows [49,50]:(12)$$V_{i}^{k+1}=\omega \cdot V_{i}^{k}+r_{1}\cdot c_{1}\cdot \left(Pbest_{i}^{k}-X_{i}^{k}\right)+r_{2}\cdot c_{2}\cdot \left(Gbest^{k}-X_{i}^{k}\right)$$
(13)$$X_{i}^{k+1}=X_{i}^{k}+V_{i}^{k+1}$$
where $V_{i}^{k}$ is the velocity of the *i*th particle at iteration *k*, ω is the inertia weight factor, $c_{1}$ and $c_{2}$ are the acceleration coefficients, $r_{1}$ and $r_{2}$ are random numbers between 0 and 1, and $X_{i}^{k}$ is the position of the *i*th particle at iteration *k*. In the velocity updating process, the parameters such as ω, $c_{1}$ and $c_{2}$ should be determined in advance, which makes it cumbersome to solve large-scale optimization problems.

As PSO is used for SVM parameter optimization, whereby the particle whose vector value can acquire the highest classification accuracy is the optimal solution. Each dataset is divided into two parts for base classifiers: training and testing. Each iteration, the training samples are used for the construction of a base classifier, then a set of parameters is generated, and the testing samples are used to prove the validity of the parameters. After that, the classification accuracy of these parameters can be achieved by comparing the predicted and original labels. The optimal classification accuracy obtained in the current generation is recorded and compared with the highest classification accuracy in the last iteration. If the former wins, the optimal solution is replaced with the current solution. When the maximum number of iterations is reached, the final parameters are obtained.

### 3.3. WSVM Ensemble

Ensemble learning trains a series of base classifiers and combines their outputs by a fusion strategy, improving the generalization performance [51]. As mentioned in Section 1, an ensemble system that uses SVMs as base classifiers is called SVM ensemble [52]. Considering the heterogeneity and homogeneity of the system [53], while reducing the running time, the bagging algorithm is adopted to aggregate the base classifiers and five kernel functions mentioned in Section 3.1 are utilized to construct SVMs. Furthermore, each kernel function is used twice and a total of 10 WSVMs are utilized for the ensemble learning.

After each WSVM has output a predicted label, it is important to adopt a fusion strategy that determines the final label of the ensemble system. In this paper, majority voting is employed as the fusion strategy. Each classifier generates a predicted label, which is taken as a vote, and the final result is determined by the vote of each classifier. For example, there is a classification problem with *n* classes (Y=1,2,…,n), a training dataset, an instance *x* to be classified and *T* classifiers. Each classifier outputs a label $L_{i}$ of *x*, where i=1,2,…,T and $L_{i}\in Y$. Let *V* be the final label of *x*, then
(14)$$V=argmax\sum_{i=1}^{T}p\left(L_{i}=y\right)$$
where y∈Y; if *a* is true, p(a)=1, otherwise p(a)=0 and *a* is the output of base classifier.

Majority voting treats the output of each classifier equally, determining the final result by counting the number of occurrences for each class. When encountering the same number of occurrences, in cases where it is difficult to decide, a random selection of these classes is adopted to determine the final result.

Following the feature extraction in Section 2, bootstrap sampling is used to generate sub-datasets from the original data, and the number of instances in each sub-dataset is not greater than that in the original data. Then, on the basis of the corresponding sub-dataset, each SVM obtains the optimal parameters by the continuous process of training and testing samples combined with PSO. After that, each base classifier predicts the labels with these parameters. Finally, majority voting is utilized to generate the final result. The schematic diagram of the proposed method is shown in Figure 1.

### 3.4. Implementation of the Proposed Method

The classification process of FWL point clouds is completed in this paper, employing WSVM ensemble. The detailed steps of the proposed method are as follows:Step 1: Acquire FWL data and filter noise in the data.Step 2: Decompose full-waveform LiDAR data and extract the features displayed in Table 1.Step 3: Use bootstrap sampling to generate sub-datasets and assign them to each base classifier.Step 4: Train base classifiers with the parameters of each particle in the population.Step 5: Obtain the classification accuracy of each particle and update the population using Equations (Equation 12) and (Equation 13).Step 6: Update global optimal accuracy and corresponding parameters.Step 6.1:Compare the classification accuracy of the particles in the current generation with the global optimal accuracy.Step 6.2:Determine whether to update the global optimal accuracy and the corresponding parameters on the basis of comparison result.Step 6.3:Return to Step 4 until the number of iterations has reached the maximum value.Step 7: Save the parameters achieved by the global optimal accuracy of each base classifier according to Steps 4–6, and take them as the parameters of each classifier.Step 8: Each base classifier predicts the labels of data with their parameters.Step 9: Output the final results with majority voting.

## 4. Experimental Results and Discussion

The experimental environment in this study was a computer with a 2.30GHz CPU and 8G of RAM. The data-processing operation was realized using MATLAB 2016a and VS2017 software. The manual classification process was accomplished using LiDAR software and visual interpretation by researchers with relevant working experience.

### 4.1. Experimental Platform and Data Information

The data used in this paper were acquired by airborne LiDAR system ALS60, in 2009, and in the form of Las 1.3. There were three study areas and the experimental data were colored according to elevation values, as shown in Figure 2a–c. Study Area 1 is flat and open, with a low point cloud density; Study Area 2 is a dense residential area; and Study Area 3 is mainly distributed with large buildings. The number of points, training and testing samples of the study areas and some other information about the study areas are shown in Table 2, and training and testing samples were generated by manual classification. To quicken the process, the number of points in each training subset generated by bootstrap sampling was one-tenth of the whole dataset. After the classification for all of the point clouds was completed, testing samples of these three study areas were used to validate the classification accuracy of the ensemble system.

### 4.2. Classification Results for Point Clouds

The classification results of the proposed classification method for FWL point clouds were compared with those of basic SVM, optimal single WSVMs, RF, ISODATA, and WSVM ensemble without parameter optimization. The classification results are shown in Figure 3, Figure 4 and Figure 5, while the CPU time of each method is shown in Table 3 (unit: s), and the classification accuracy of each method is shown in Table 4. The ground objects were classified into low vegetations (blue color), trees (green color), buildings (yellow color) and others (red color). Low vegetations includes grass and crops, while others includes mainly artificial surfaces such as roads and concrete floors.

As shown in Table 3, the CPU time of WSVM ensemble without parameter optimization was lower than any other compared methods, and those of RF and optimal WSVMs were relatively high. The CPU time of the proposed method was 85.0167, 78.4931 and 97.0145 s, respectively, which was significantly reduced when compared to optimal single WSVM. As shown in Table 4, the proposed method performed better than RF and ISODATA; it also obviously outperformed basic SVM and WSVM ensemble without parameter optimization, which illustrates the importance of SVM kernel function and parameter optimization. In addition, the classification accuracy of optimal single WSVMs proved the applicability of WSVM in such classification problems. Finally, from the perspective of the ensemble, the classification accuracy of WSVM ensemble was higher than optimal single WSVMs used in the ensemble system. The classification accuracies via optimal single WSVMs of the three study areas were 96.5720%, 94.5937% and 92.1143%, which were 1.1757%, 0.5018% and 1.7179% lower, respectively, than those of the proposed method. Considering its CPU time, as presented in Table 3, this method has good application prospect.

As shown in Figure 3, Figure 4 and Figure 5, the results of ISODATA and basic SVM had obvious misclassifications, which proved that unsupervised classification and improper kernel functions of SVM have difficulty distinguishing the objects with approximate feature values. It was also found that the results of WSVM ensemble without parameter optimization had some noise (see, for example, left center of Figure 3e and the left half of Figure 5e), which illustrates that the random parameters may be not suitable in such classification problems. Optimal single WSVMs and the proposed method had similar results, but further comparison of these two methods’ results revealed that the latter method had better classification ability for details. As shown in Figure 3, the road in the lower left corner extracted by the proposed method is more complete than optimal single WSVM. As shown in Figure 5, the low vegetations in the lower left corner are misclassified as others by optimal single WSVM, while these errors do not exist in the result of the proposed method.

## 5. Conclusions

A FWL point cloud classification method employing WSVM ensemble is proposed in this paper, which utilizes five different wavelet kernel functions and ensemble learning. In the process of classification, geometric and waveform features are adopted, and bagging algorithm is used to integrate the base classifiers. Further, in each base classifier, PSO is employed to optimize WSVM parameters to obtain satisfactory classification results. The classification accuracy of all study areas was over 93%, which proved the feasibility of the proposed method. In addition, it was higher than that of basic SVM, optimal single WSVMs, RF, ISODATA and WSVM ensemble without parameter optimization. Moreover, in the case of a similar classification accuracy, the CPU time of the WSVM ensemble was less than single WSVMs. In conclusion, the proposed method can acquire accurate and effective results of FWL point cloud classification and has strong application potential for large-scale classification problems. On the basis of this work, future research such as the optimization of FWL feature extraction will be conducted. 

## Figures and Tables

**Figure 1 sensors-19-03191-f001:**
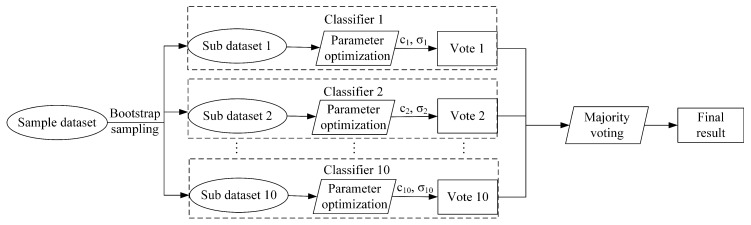
Schematic diagram of the proposed method.

**Figure 2 sensors-19-03191-f002:**
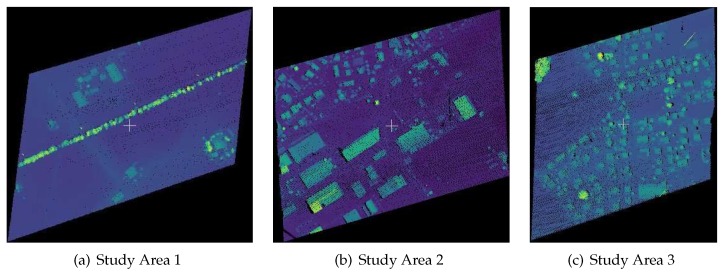
Original data of the three study areas.

**Figure 3 sensors-19-03191-f003:**
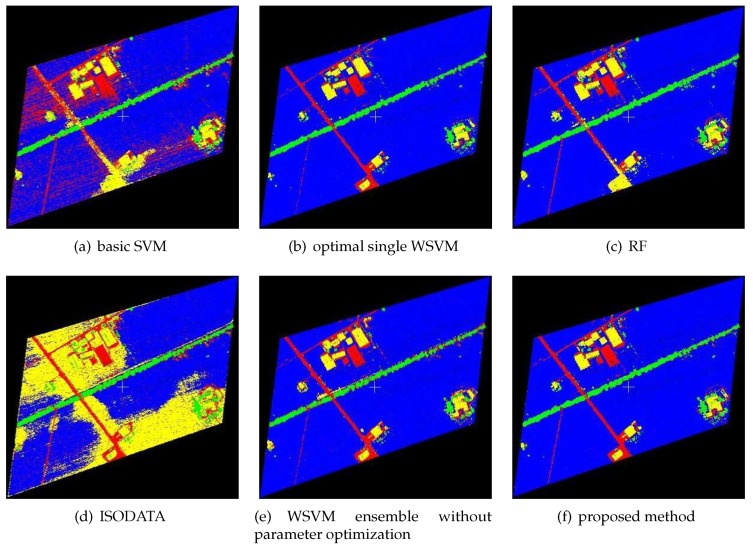
Classification results of Study Area 1.

**Figure 4 sensors-19-03191-f004:**
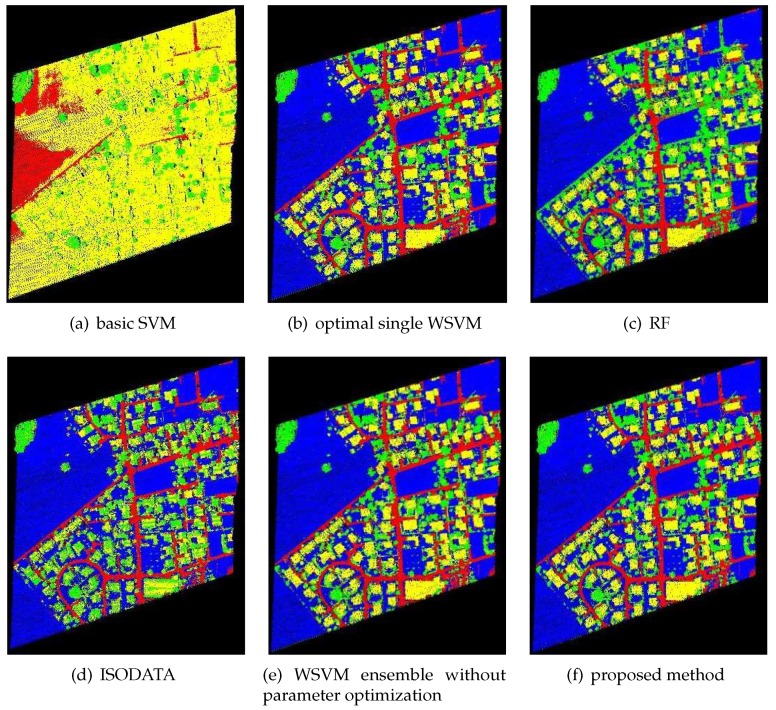
Classification results of Study Area 2.

**Figure 5 sensors-19-03191-f005:**
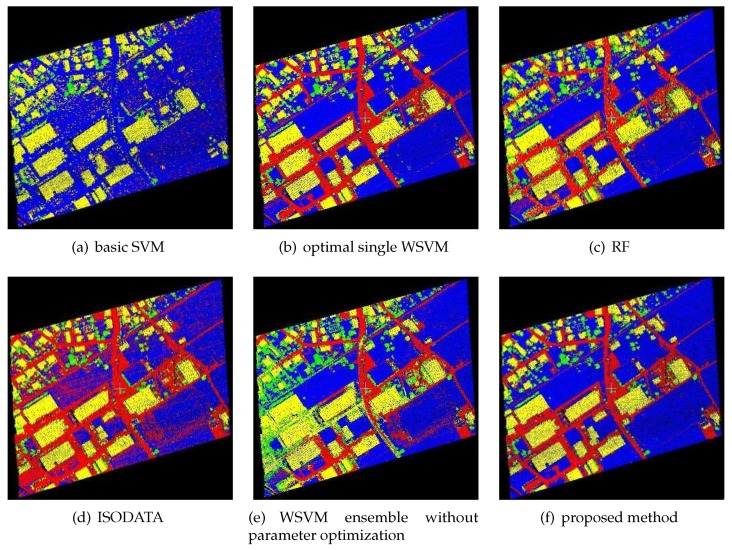
Classification results of Study Area 3.

**Table 1 sensors-19-03191-t001:** Features for classification.

Feature Type	Feature Name	Formula	Explanation
Geometric	*h*	/	In general, elevation can effectively distinguish between ground and off-ground points, but, for trees, houses, and hillsides with similar elevations, absolute elevation may not work.
	$\sigma_{h}$	$\sigma_{h}=\sqrt{\frac{\sum_{i=1}^{n}{\left(Z-Z_{ave}\right)}^{2}}{n-1}}$	*Z* denotes the current point, $Z_{ave}$ denotes the average elevation of all points in ω, and *n* is the number of points in ω. High vegetation and the edges of buildings often have a greater height difference.
	ρ	$\rho =n/v_{\omega }$	$v_{\omega }$ is the volume of ω. Generally speaking, ρ of building walls and trees is lower than others.
	*c*	$c=\lambda_{1}/\left(\lambda_{1}+\lambda_{2}+\lambda_{3}\right)$	*c* reflects the shape of the surface of the object, and the canopy usually has a high value.
	$N_{Z}$	/	Deviation angle of a normal vector from the vertical direction, reflecting the flatness of the ground object.
	$\sigma_{N_{Z}}$	$\sigma_{N_{Z}}=\frac{\sqrt{N_{X}^{2}+N_{Y}^{2}}}{N_{Z}}$	The variance of the vertical angles of 3D points in ω, reflecting the shape of the ground object.
Waveform	*A*	/	The value of the natural surface and building is the highest, and that of the asphalt surface and trees is low, thus it can distinguish between vegetation and artificial objects.
	σ	/	σ reflects the time that the laser pulse interacts with the ground object. Due to the scattering effect of the canopy on the laser, σ can distinguish between non-vegetation and vegetation.
	*u*	/	*u* can be used to calculate the distance between the laser emission location and the target.
	*I*	/	*I* is the amount of energy returned by the laser pulse interacting with the ground objects whose characteristic is similar to *A*.

**Table 2 sensors-19-03191-t002:** Experimental data information.

Experimental	Data Area	Total	Training	Testing	Point Cloud
Data	$\left(m^{2}\right)$	Points	Samples	Samples	Density
Study Area 1	203,833	227,078	6450	5683	1.11
Study Area 2	180,030	283,315	6489	5070	1.57
Study Area 3	131,767	226,123	6739	5399	1.72

**Table 3 sensors-19-03191-t003:** CPU time of different methods (s).

Experimental	Basic	Optimal	RF	ISODATA	Non-Optimization	Proposed
Data	SVM	WSVM				Method
Study Area 1	78.6326	123.0621	139.9527	82.2098	69.6642	85.0167
Study Area 2	73.5228	111.6443	123.8943	76.3754	66.0029	78.4931
Study Area 3	88.9489	130.5389	152.0112	93.0824	77.4131	97.0145

**Table 4 sensors-19-03191-t004:** Classification accuracy of different methods (%).

Experimental	Basic	Optimal Single	RF	ISODATA	Non-Optimization	Proposed
Data	SVM	WSVMs				Method
Study Area 1	66.0743	96.5720	94.8236	55.7943	93.4956	97.7477
Study Area 2	55.5702	94.5937	88.2017	77.6837	90.3324	95.0955
Study Area 3	66.8505	92.1143	93.1469	75.7884	84.4282	93.8322

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
