# Peer review of "Full-Waveform LiDAR Point Clouds Classification Based on Wavelet Support Vector Machine and Ensemble Learning"

_sensors, 2019, doi:10.3390/s19143191_

Round 1

Reviewer 1 Report

The a manuscript is well written with sufficient scientific content. 

I recommend the publication of this manuscript in the current form. 

Author Response

Thank you for your recognition and hard work on this manuscript.This will inspire us to work harder.

Reviewer 2 Report

The article presents the use of Support Vector Machine and Ensemble Learning to classify a point cloud.

Presenting the case study and point cloud description, the authors showed the complexity of the problem, however they did not explain it directly in case of using the data from different LiDAR sensors, i.e. whether they used terrestrial, mobile or airborne system and whether it affects the final result.

In the following parts of the manuscript, the authors presented transformations of the wavelet function as the main mathematical methods. From my experience, during the classification, the data filtration is important, where the smallest percent filtration errors are obtained in the case of an infinitely high threshold elimination value. Due to the fact that I did not find it in the text directly, I have questions to authors: Did filtration have an impact on the final result and How did the authors check it? I am particularly interested in the extraction of data written by the authors in the 201 line.

In my opinion, the problem may be up-to-date, the more the more often we can see the use of the fusion of various sensors.

Reviewer 3 Report

This paper aims to complete full-waveform LiDAR point clouds classification by wavelet support vector machine and ensemble learning, employing five wavelet kernel functions. At the same time, particle swarm optimization (PSO) is utilized to optimize the SVM parameters. The subject of the paper is interesting but it is not clearly presented.  The authors should improve the text through the manuscript to make clear for the readers. For example the abstract and the introduction should be improved and to be more specific according to the subject of the paper.

Some observations:

L.1- Light Detection and Ranging (LiDAR), as a part of remote sensing, can generate three-dimensional (3D) point clouds and has been used to conduct classification in many cases. --- This sentence should be improved: Light Detection and Ranging (LiDAR) can generate three-dimensional (3D) point clouds and has been used to conduct classification in many cases ?.

L.3 - Support vector machine (SVM) is one of the commonly used classifiers as the increase of data dimension ?.

L.80 - The remainder of this paper is organized as follows. Section 2 presents the extracted features of full-waveform LiDAR point clouds. The methodology of LiDAR point clouds classification is illustrated in section 3, including the construction of wavelet kernel functions, parameter optimization of SVM via PSO, and the WSVM ensemble system. Section 4 presents the experimental results and discussion, providing a comparison and analysis of the experiment results. Finally, section 5 concludes the paper and proposes future research. ---  This paragraph can be removed from the text.

And so on.

Round 2

Reviewer 3 Report

This paper aims to complete full-waveform LiDAR point clouds classification by wavelet support vector machine and ensemble learning, employing five wavelet kernel functions. At the same time, particle swarm optimization (PSO) is utilized to optimize the SVM parameters. The subject of the paper is interesting and the text has been improved in this second version. The authors improved the text through the manuscript making it clear for the readers. The authors followed the reviewers comments.

Author Response

This version further refines the English expression . Here, we would like to thank the reviewer for the valuable comments, and for your time and hard work.